# TRPS1 modulates chromatin accessibility to regulate estrogen receptor alpha (ER) binding and ER target gene expression in luminal breast cancer cells

**Thomas G. Scott[1], Kizhakke Mattada Sathyan[2,3], Daniel Gioeli[4,5], Michael J. Guertin** [2,3] *

**1** Department of Biochemistry and Molecular Genetics, University of Virginia, Charlottesville, Virginia, United States of America, **2** Center for Cell Analysis and Modeling, University of Connecticut, Farmington, Connecticut, United States of America, **3** Department of Genetics and Genome Sciences, University of Connecticut, Farmington, Connecticut, United States of America, **4** Department of Microbiology, Immunology, and Cancer, University of Virginia, Charlottesville, Virginia, United States of America, **5** Cancer Center Member, University of Virginia, Charlottesville, Virginia, United States of America

* guertin@uchc.edu

**Data Availability Statement:** All analysis details and code are available at https://guertinlab.github.io/TRPS1_ER_analysis/Vignette.html. Raw sequencing files and processed counts and bigWig

## Abstract

Common genetic variants in the repressive GATA-family transcription factor (TF) *TRPS1* locus are associated with breast cancer risk, and luminal breast cancer cell lines are particularly sensitive to *TRPS1* knockout. We introduced an inducible degron tag into the native *TRPS1* locus within a luminal breast cancer cell line to identify the direct targets of TRPS1 and determine how TRPS1 mechanistically regulates gene expression. We acutely deplete over 80 percent of TRPS1 from chromatin within 30 minutes of inducing degradation. We find that TRPS1 regulates transcription of hundreds of genes, including those related to estrogen signaling. TRPS1 directly regulates chromatin structure, which causes estrogen receptor alpha (ER) to redistribute in the genome. ER redistribution leads to both repression and activation of dozens of ER target genes. Downstream from these primary effects, TRPS1 depletion represses cell cycle-related gene sets and reduces cell doubling rate. Finally, we show that high TRPS1 activity, calculated using a gene expression signature defined by primary TRPS1-regulated genes, is associated with worse breast cancer patient prognosis. Taken together, these data suggest a model in which TRPS1 modulates the genomic distribution of ER, both activating and repressing transcription of genes related to cancer cell fitness.

## Author summary

Breast cancer is the most common cancer among women. The majority of cases are luminal, which tend to be estrogen receptor alpha (ER)-positive. ER is well-studied among transcription factors (TFs) because it is ligand-activated. This allows for the rapid induction of ER activity and the identification of primary estrogen-responsive genes. Most TFs have not been so extensively characterized, because their activity is not so rapidly

files are available from GEO SuperSeries accession record GSE236176, with SubSeries accession records GSE236175 (ATAC-seq), GSE236174 (ChIP-seq), GSE236172 (time course PRO-seq), and GSE251772 (three clone PRO-seq).

**Funding:** This work was funded by R35-GM128635 to MJG. 5T32GM007267-39 and 5T32GM008136-35 supported TGS. The funders had no role in study design, data collection and analysis, decision to publish, or preparation of the manuscript.

**Competing interests:** The authors declare no competing interests.

perturbable. TRPS1 is an atypical GATA family TF that is associated with corepressor complexes and transcriptional repression. Here, we use an inducible degron tag system to rapidly deplete endogenously-tagged TRPS1 in luminal breast cancer cells within 30 minutes. We find that TRPS1 directly decreases local chromatin accessibility. This decreases ER binding intensity at TRPS1-proximal ER binding sites. As an indirect effect, ER binding intensity distal to TRPS1 increases in intensity. These effects on ER binding are associated with changes in ER target gene transcription, repressing or activating these genes in concordance with the effect on ER binding intensity. Our work is consistent with a model in which TFs either exclusively activate or exclusively repress transcription of their direct target genes, with indirect primary response genes changing due to the redistribution of limiting activating TFs or coactivators.

## Introduction

Breast cancer is the most frequently diagnosed cancer in women, with an estimated lifetime risk of about 1 in 8 for women in the United States [1]. Far from a monolithic disease, breast tumors can be classified into subtypes based on gene expression, histology, and immunohistochemistry [2, 3]. The most common subtype is luminal breast cancer, which is typically estrogen receptor alpha (ER)-positive [4].

High lifetime exposure to endogenous estrogen is a strong risk factor for breast cancer incidence [5]. Estrogen is a potent hormone that binds to ER, a ligand-activated transcription factor (TF), which then homodimerizes, binds to reverse palindromic pairs of AGGTCA motifs on DNA, and recruits cofactors to activate hundreds of genes that promote cell growth and proliferation [6–8]. In additional to surgery, radiation, and traditional chemotherapy, endocrine therapies that inhibit endogenous estrogen production or binding to ER provide a significant survival benefit to luminal breast cancer patients [4, 9, 10]. However, patients with advanced disease frequently develop resistance to these therapies, though many endocrine therapy-resistant luminal tumors still remain dependent on ER activity [11]. Thus, there is a need to identify additional factors that regulate ER activity or genomic binding and contribute to breast tumor progression.

TRPS1 is a member of the GATA-family of TFs that bind to (A/T)GATA(A/G) motifs on DNA [12]. In contrast to the other six members of the GATA family that activate transcription, TRPS1 directly represses transcription of target genes via its unique IKZF1-like zinc fingers [13]. TRPS1 has been shown to interact with multiple corepressors and lysine deacetylases, including members of the NuRD and coREST complexes, to regulate transcription [14–18].

*TRPS1* was first described as the gene mutated in cases of tricho-rhino-phalangeal syndrome, an autosomal dominant disorder characterized by developmental abnormalities of the hair, nose, and fingers [19]. *TRPS1* is crucial for the proper development of several tissues, including hair, bone, and kidney [20, 21]. As with many developmentally important genes co-opted during the process of cancer initiation and progression, *TRPS1* is commonly overexpressed in breast tumors, both relative to normal tissue and relative to other tumor types [22, 23].

The transcriptional program that TRPS1 regulates in breast cancer is not fully understood. Knockdown of *TRPS1* in various breast cancer cell lines has been shown to increase markers of epithelial to mesenchymal transition and genome instability [24–27]. Additionally, TRPS1 binding sites on chromatin overlap with those of YAP1 and ER, though this is coupled with a genome-wide activation of YAP1 target genes but repression of ER target genes [16, 17].

A key feature of these previous studies is the use of extended knockdown kinetics with traditional RNA interference methods. Days after knockdown, the resultant effects represent not only the primary TRPS1-responsive genes but also secondary and compensatory effects. As such, we do not know which genes TRPS1 directly regulates and whether these genes are important for breast cancer cell growth and proliferation.

In this study, we set out to directly assay the primary effects of TRPS1 on chromatin accessibility, ER binding, and transcription in luminal breast cancer cells. To do so, we acutely depleted TRPS1 protein levels using an inducible degron tag inserted into the endogenous *TRPS1* locus. By performing sensitive genome-wide assays minutes to hours after TRPS1 depletion, we demonstrated that TRPS1 changes chromatin structure, which allows ER to redistribute in the genome. Along with this redistribution, we propose that TRPS1 both directly represses and indirectly activates dozens of ER target genes at baseline. Furthermore, we defined a signature of primary TRPS1-regulated genes that predicts breast cancer patient prognosis.

## Results

### *TRPS1* is associated with breast cancer incidence and promotes breast cancer cell number accumulation

A recent genome-wide association study (GWAS) identified 32 novel single nucleotide polymorphisms (SNPs) associated with breast cancer susceptibility [28]. When we queried the NHGRI EBI GWAS Catalog to find published associations with genetic variants within the *TRPS1* genomic locus, we found one of the lead SNPs from this study [29]. Furthermore, in the authors' subtype-specific analysis of the results, the association with this variant was strongest for luminal breast cancers relative to other subtypes [28].

We used LocusZoom to plot the data within this locus (Fig 1A). A plot of these data indicates that two sets of SNPs have low linkage disequilibrium with one another, indicating that they are inherited independently and each confer risk. One set of variants is within an intronic region of the *TRPS1* gene, and one is about 400 kilobases upstream from the transcription start site (TSS) of the *TRPS1* gene. There are often many genes in close proximity to the lead SNP in a GWAS, making it difficult to predict which gene mediates the effect on the associated phenotype. However, in this case the nearest gene is almost a megabase upstream from *TRPS1*, suggesting that *TRPS1* contributes to the breast cancer susceptibility associated with one or both of these sets of genetic variants.

Based on this result, we hypothesized that perturbation of *TRPS1* in luminal breast cancer cell lines would affect cell fitness. Using data from the Cancer Dependency Map project, we found that sensitivity to *TRPS1* knockout correlated with sensitivity to knockout of *ESR1*, the gene encoding ER (Fig 1B) [31]. Furthermore, while *TRPS1* knockout led to an increase in cell number for most cancer cell lines, luminal breast cancer cell lines were significantly enriched for *TRPS1* dependency (Fig 1C).

Taken together, these data indicate that *TRPS1* influences breast cancer incidence and is required for maximal breast cancer cell fitness. Next we sought to determine how TRPS1 regulates its target genes to mediate these breast cancer patient and cellular outcomes.

### Endogenously degron-tagged TRPS1 is rapidly degraded in T47D cells

To rapidly deplete TRPS1 and isolate primary TRPS1-regulated genes, we employed the dTAG system for targeted protein degradation [32, 33]. We inserted an inducible degron tag into the endogenous *TRPS1* locus in the luminal breast cancer cell line T47D. We generated three

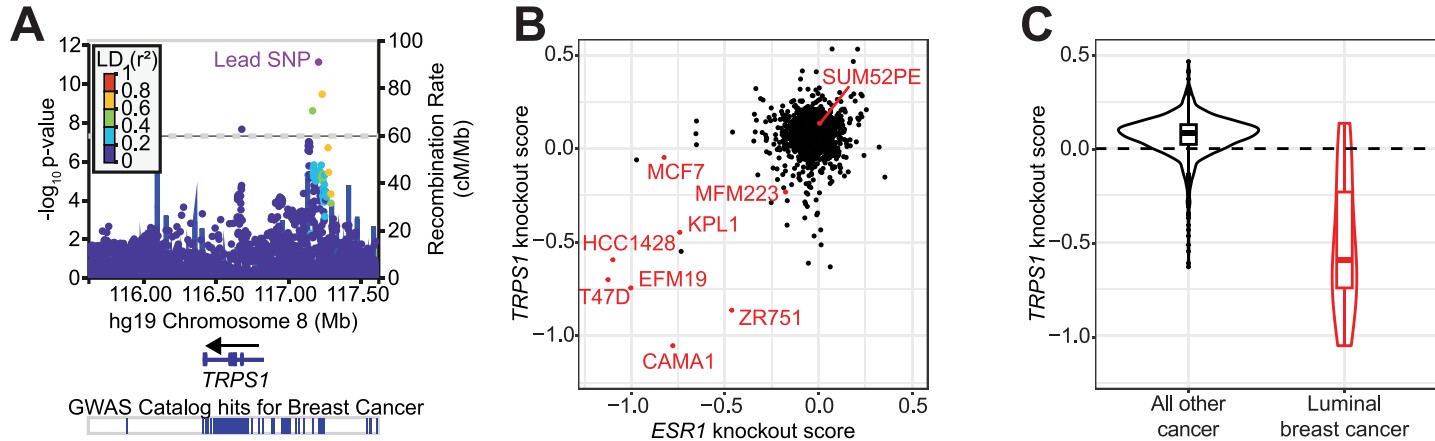

**Fig 1. *TRPS1* is associated with breast cancer incidence and promotes breast cancer cell fitness.** A: LocusZoom plot of the *TRPS1* genomic locus depicting the location and significance of SNPs associated with breast cancer susceptibility. *TRPS1* is the closest gene to two sets of genetic variants in low linkage disequilibrium with one another. Data from [28], generated with summary statistics downloaded from the NHGRI-EBI GWAS Catalog, using LocusZoom [29, 30]. B: Scatter plot of *TRPS1* and ESR1 knockout scores for each gene tested. Scores are normalized such that knockout of a gene with a score of 0 has no effect on cell number, and knockout of a gene with a score of -1 has an effect equal to that of knocking out one of a set of universally essential genes. Luminal breast cancer cell lines are colored in red. The data are from the Cancer Dependency Map project [31]. C) Violin and box and whisker plots of *TRPS1* knockout scores from (B) for luminal breast cancer cell lines versus all other cell lines. Wilcoxon rank sum test p-value of $3.2*10^{-5}$.

independent clones that express the tagged TRPS1 protein that can be degraded by the addition of the small molecules dTAG-13 and dTAG$^V$-1 at 50nM each (dTAG) (Fig 2A). Of note, we depleted around 50% of TRPS1 from whole cell lysates in 10 minutes of treatment with dTAG, as determined by quantitative western blot, with less than 10% detected as soon as 20 minutes and as late as 48 hours after treatment (Fig 2B).

To ensure this treatment depleted TRPS1 from chromatin, we performed chromatin immunoprecipitation with sequencing (ChIP-seq) using an anti-HA antibody to recognize the 2xHA tag within the degron tag. Our ChIP-seq libraries were of high quality, as measured using the quality control metrics of fraction of reads in peaks (S1 Fig) and the peak calling-

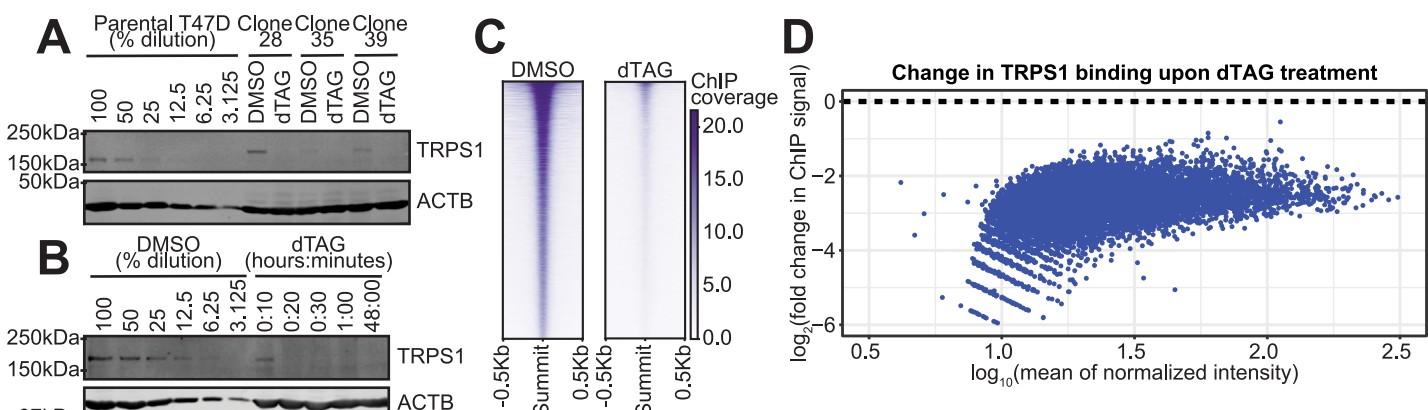

**Fig 2. Endogenously degron-tagged TRPS1 is rapidly degraded in T47D cells.** A: Quantitative Western blot with a serial dilution of the parental T47D cells followed by three independent dTAG-TRPS1 clones treated with DMSO or dTAG-13 and dTAG$^V$-1 at 50nM each (dTAG) for 2 hours. Membranes were probed with anti-TRPS1 or anti-ACTB antibodies. B: Quantitative Western blot with a serial dilution of dTAG-TRPS1 Clone 28 followed by a time course of treatment with dTAG. Membranes were probed as in (A). C: Heatmap of TRPS1 ChIP-seq peaks, in rows ranked by intensity, in cells treated with DMSO or dTAG for 30 minutes. D: MA plot of TRPS1 ChIP-seq peaks, with fold change values representing binding intensity in the dTAG condition relative to the DMSO condition. All points are colored blue to indicate they are significantly decreased at an FDR of 0.1.

independent strand cross-correlation (S2 Fig), generated using the ChIPQC R package [34]. We called peaks using MACS2 [35], using all DMSO samples together and all dTAG samples as the control. We observed a genome-wide decrease in TRPS1 binding intensity, with over 80 percent of TRPS1 depleted from chromatin after 30 minutes of dTAG treatment (Fig 2C and 2D).

## TRPS1 directly represses regulatory element activity

With this system in hand, we set out to test the effects of TRPS1 depletion on regulatory element (RE) (i.e., enhancer or promoter) activity. To capture the dynamics of chromatin accessibility after TRPS1 depletion, we conducted a time course analysis using the Assay for Transposase-Accessible Chromatin with sequencing (ATAC-seq). The time points included dTAG treatments at 30 minutes, 1 hour, 2 hours, 4 hours, and 24 hours, while a DMSO treatment served as the vehicle control, assigned as the zero minute time point. Our ATAC-seq libraries were of high quality, as determined by plotting the distribution of fragment sizes (S3 Fig) and the enrichment of signal around TSS's (S4 Fig), generated using the ATACseqQC R package [36]. We generated a consensus peak set using MACS2 [35] for all samples together. At the earliest time point after degradation, our best estimate of the primary effects of TRPS1 depletion, we identify 472 peaks that increased in intensity and 36 that decreased in intensity, at a false discovery rate (FDR) of 0.1. (Fig 3A and S1 and S2 Tables). We hypothesized that the increased peaks result from loss of a direct TRPS1 reduction of chromatin accessibility, with the decreased peaks an indirect effect of the redistribution of limiting cofactors.

To test this hypothesis, we performed *de novo* motif identification in the increased and decreased peaks. We identified a GATA motif in the increased peaks but not the decreased peaks (Fig 3B). To explicitly calculate the motif prevalence in each class of peaks, we found individual motif occurrences genome-wide and intersected the peaks with these motif instances. We found a significant enrichment of a representative GATA motif in the increased peaks over the unchanged and decreased peaks (Fig 3C).

To identify additional TFs that might be contributing to changes in chromatin accessibility, we analyzed motif enrichment in both increased peaks relative to unchanged peaks as well as decreased peaks relative to unchanged peaks (S3 and S4 Tables). To our surprise, we identified nuclear receptor motifs in both classes of peaks. To follow up this analysis, we again calculated motif prevalence in each class of peaks for a representative ER half-site. For this motif, we found a significant enrichment in both increased and decreased peaks relative to unchanged peaks (Fig 3D).

We predicted that the enrichment of the GATA motif within increased peaks would wane over time. To test this prediction, we turned to our time course data. We identified ATAC-seq peaks that were significantly changed over the time course at an FDR of 0.1, using a likelihood ratio test within DESeq2 [37] to identify peaks for which including a variable for the time point increased the predictive power of the model over one without this information. We performed hierarchical clustering of these peaks and found that the replicates for each time point clustered together and that the majority of dynamic peaks changed gradually over the time course, with additional clusters displaying different kinetics (Fig 3E). We called increased, unchanged, and decreased peaks as in Fig 3A for each time point relative to the control condition. We then calculated GATA motif prevalence in each set of peaks as in Fig 3C. As we expected, the GATA motif prevalence in the increased peaks decreased over time, but the majority of increased peaks at 24 hours still contained a GATA motif (Fig 3F). This is consistent with the primary effects of TRPS1 depletion driving a large proportion of the changes in chromatin accessibility even as late as 24 hours after TRPS1 depletion.

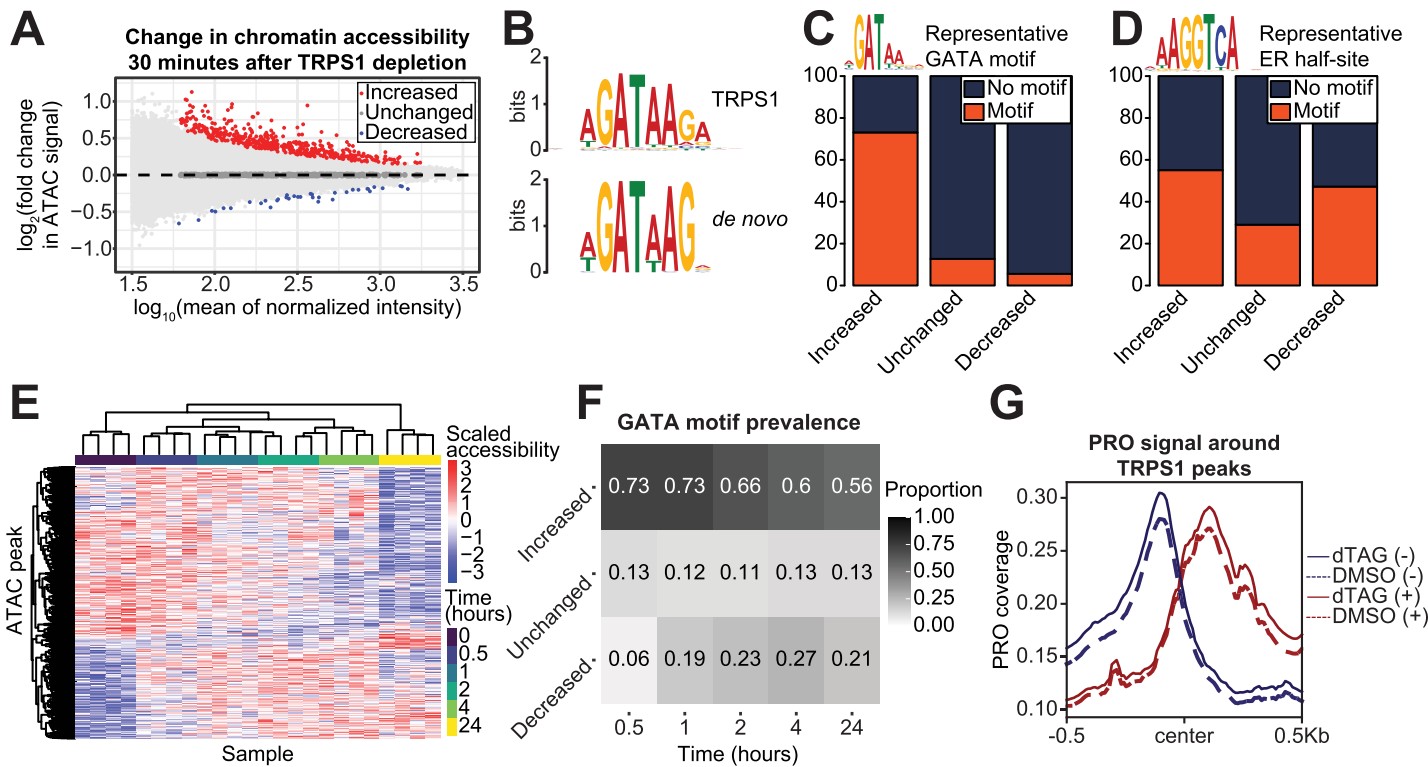

**Fig 3. TRPS1 directly represses regulatory element activity.** A: MA plot of ATAC-seq peaks, with fold change values representing accessibility in the 30 minute dTAG-13 and dTAG$^V$-1 at 50nM each (dTAG) treatment condition relative to the DMSO condition. B: *De novo* motif identified in increased peaks from (A) (below), matched to the TRPS1 motif (above). C: Bar charts of prevalence of a representative GATA motif in increased, unchanged, and decreased peaks from (A). Chi-square test p-value $< 2.2*10^{-16}$. D: Bar charts of prevalence of a representative ER half-site in increased, unchanged, and decreased peaks from (A). Chi-square test p-value $4.5*10^{-15}$. E: Heat map with hierarchical clustering of chromatin accessibility in ATAC-seq peaks that are significantly changed over the time course at an FDR of 0.1. F: Heat map of prevalence of the representative GATA motif in increased, unchanged, and decreased peaks, as in (C), for each time point relative to the DMSO condition. G: Density plot for composite PRO-seq signal across all TRPS1 ChIP-seq peaks, separated by strand and treatment condition.

As an orthogonal readout of RE activity, we measured bidirectional transcription around TRPS1 binding sites using precision genomic run-on with sequencing (PRO-seq) [38]. Our libraries were of high quality using several quality control metrics (S5 Fig) [39]. Using a window centered on each summit of TRPS1 ChIP-seq intensity, we observed an increase in bidirectional transcription 30 minutes after TRPS1 depletion (Fig 3G).

Since composite profiles have limitations, we chose to look more closely at each TRPS1 peak and determine if increasing bidirectional transcription is a reproducible trend. We counted PRO reads in the window around the TRPS1 peak summits shown in Fig 3G and used DESeq2 to identify differentially transcribed regions. We normalized our counts based on the size factors generated from the DESeq2 analysis of the reads in genes from Fig 4. As read counts in small putative regulatory elements tend to be fewer than in large genes, it is not surprising that only 9 of these regions have individually statistically significantly higher bidirectional transcription and none with significantly lower transcription, at a false discovery rate of 0.1 (S6 Fig). However, we did find a TRPS1 cistrome-wide increase in PRO signal, with over 62% of these regions increasing in bidirectional transcription upon TRPS1 depletion. We performed ANOVA on a linear model predicting the logarithm of the normalized PRO reads for each region based on the DMSO or 30 minute dTAG treatment condition across the four replicates, and the p-value for the F-test was $< 2.2*10^{-16}$.

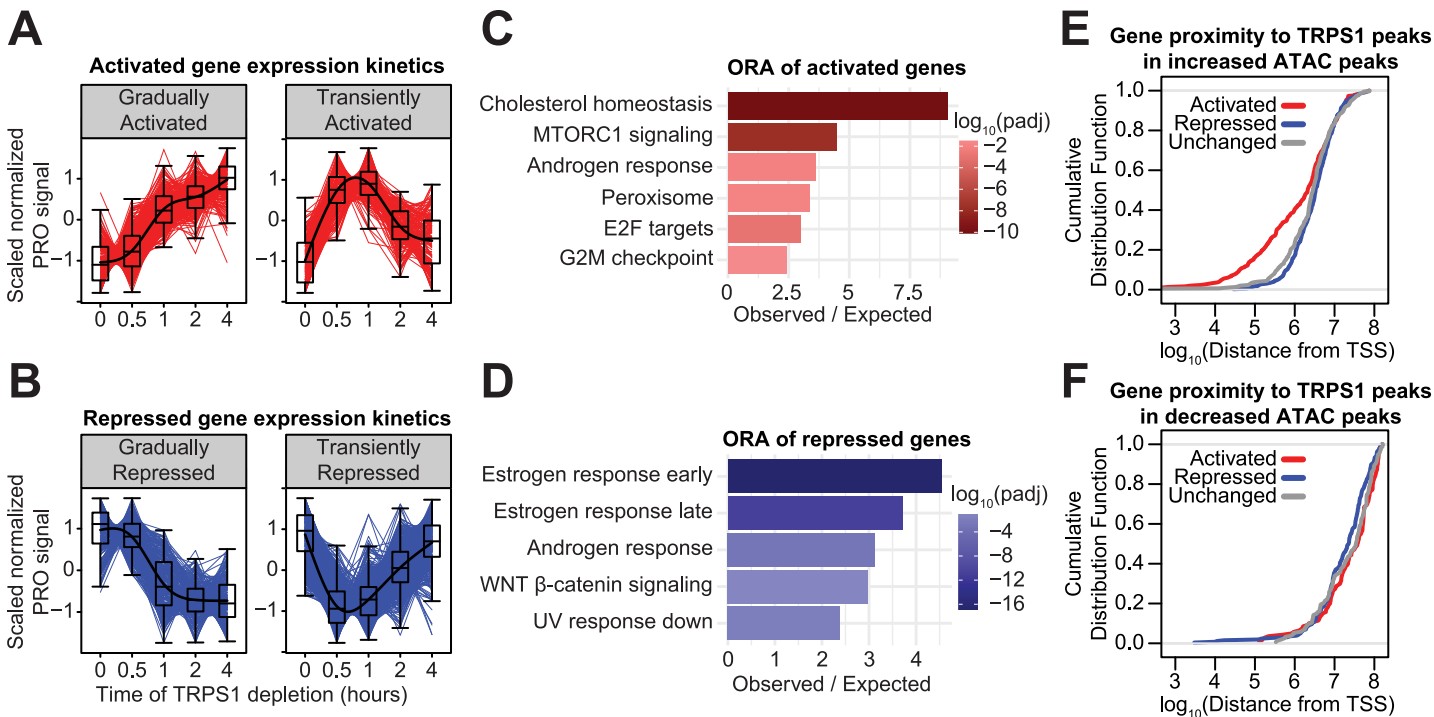

**Fig 4. TRPS1 directly represses transcription of target genes.** A: Kinetic traces of the two major clusters of activated genes over the time course. B: Kinetic traces of the two major clusters of repressed genes over the time course. C: Over-representation analysis of the activated genes from (A). D: Over-representation analysis of the repressed genes from (B). E: Cumulative distribution function plot of the distance from the TSS of each gene to the nearest TRPS1 ChIP-seq peak overlapping an increased ATAC-seq peak, by gene class. Kolmogorov–Smirnov test between activated and unchanged genes: p-value = 0.011. F: Cumulative distribution function plot of the distance from the TSS of each gene to the nearest TRPS1 ChIP-seq peak overlapping an decreased ATAC-seq peak, by gene class. KS test between repressed and unchanged genes: p-value > 0.1.

The biological interpretation is consistent with our other results that indicate a role of TRPS1 in chromatin compaction. We would not necessarily expect an increase in bidirectional transcription at each TRPS1 peak upon TRPS1 degradation, but the increase in chromatin accessibility could facilitate downstream factors that promote transcription initiation. As chromatin accessibility and bidirectional transcription can affect one another, we next isolated ATAC-seq peaks without bidirectional transcription, as identified using dREG [40]. As we saw in Fig 3A, chromatin accessibility predominantly increased upon TRPS1 depletion (S7 Fig). Along with our accessibility data and motif analysis, these data indicate that TRPS1 directly represses RE activity primarily via its effect on chromatin accessibility.

## TRPS1 directly represses transcription of target genes

Downstream from the changes in RE activity, we measured changes in nascent transcription within genes with PRO-seq over the same time course of TRPS1 depletion as in the ATAC-seq experiment. As with our ATAC-seq time course analysis, we identified genes that were significantly changed over the time course at an FDR of 0.1, using a likelihood ratio test within DESeq2 [37] to identify genes for which including a variable for the time point increased the predictive power of the model over one without this information. We identified 1,425 dynamic genes over the time course and performed hierarchical clustering to classify the genes based on their expression kinetics (Fig 4A and 4B and S5, S6, S7 and S8 Tables). Over-representation analysis (ORA) of the activated genes identified several enriched Hallmark gene sets [41], most

prominently cholesterol homeostasis genes (Fig 4C). ORA on the repressed genes revealed that the two estrogen response gene sets were the most significantly enriched of the Hallmark gene sets.

We hypothesized that TRPS1 regulates these gene sets by distinct mechanisms. Specifically, we predicted that TRPS1 directly represses the dTAG-activated genes by repressing the activity of REs proximal to these genes. To test this prediction, we measured the distance from the transcription start site (TSS) of each gene to the nearest TRPS1 ChIP-seq peak overlapping an increased ATAC-seq peak. By constructing a cumulative distribution function, we found that the activated genes are significantly closer to these activated REs (Fig 4E).

In contrast to the genes activated by TRPS1 depletion, we predicted that the effects on the genes repressed by TRPS1 depletion are indirect and distal to TRPS1 binding. To test whether TRPS1 directly activates a subset of REs to activate transcription of proximal genes, we measured the distance from the TSS of each gene to the nearest TRPS1 ChIP-seq peak overlapping a decreased ATAC-seq peak. There are few examples of this class of RE, so the distances were much farther, and there was no significant enrichment of repressed genes proximal to these peaks (Fig 4F). These data suggest that, while TRPS1 positively and negatively regulates hundreds of primary response genes, TRPS1 only represses transcription of its direct target genes.

## TRPS1 redistributes ER binding to modulate ER target gene transcription

Based on the correlation between cancer cell line sensitivity to *TRPS1* knockout and *ESR1* knockout (Fig 1B), the enrichment of an ER binding motif in both increased and decreased ATAC-seq peaks (Fig 3D), and the over-representation of estrogen response gene sets in the genes repressed by TRPS1 depletion (Fig 4D), we focused on ER target genes to explore a possible mechanism by which TRPS1 indirectly activates transcription. We first defined direct ER target genes using our previously-generated PRO-seq data from parental T47D cells that were hormone starved for three days and then acutely stimulated with estrogen or a DMSO vehicle control for 90 minutes (S8 Fig) [39]. We exclusively focused on estrogen-activated genes because ER directly activates these genes [8, 42, 43]. 65 ER target genes were activated, and 58 were repressed by acute TRPS1 depletion (Fig 5A). To test the robustness of this change in ER target gene transcription, we additionally performed PRO-seq in each of the three independent TRPS1-dTAG clones generated from the parental T47D cells. Indeed, the genes we identified as TRPS1-regulated ER target genes in the one clone used for the time course experiment tend to be regulated in the same direction upon TRPS1 degradation across these three clones, suggesting that this effect is robust across the cell lines in which we can acutely deplete TRPS1 (S9 Fig).

We hypothesized that these changes in ER target gene transcription are mediated by changes in the genomic distribution of ER binding. We performed ER ChIP-seq to test the prediction that ER binding intensity proximal to dynamic ER target genes would change in concordance with the change in gene transcription. As before, our ChIP-seq libraries were of high quality (S1 and S2 Figs). We called peaks using MACS2 [35], using all ER samples together and all IgG samples as the control. Consistent with our hypothesis, we found that ER ChIP-seq peaks within a 100 kilobase (kb) window around the TSS of activated genes tended to increase in intensity, and ER ChIP-seq peaks within a 100kb window around the TSS of repressed genes tended to decrease in intensity (Fig 5B).

We further hypothesized that only the increased ER binding sites represent a direct effect of TRPS1 activity. Consistent with this hypothesis, we found that ER binding proximal to TRPS1 tends to increase in intensity, and ER binding distal to TRPS1 tends to decrease in intensity (Fig 5C). Together, these data suggest a model in which TRPS1 depletion redistributes ER

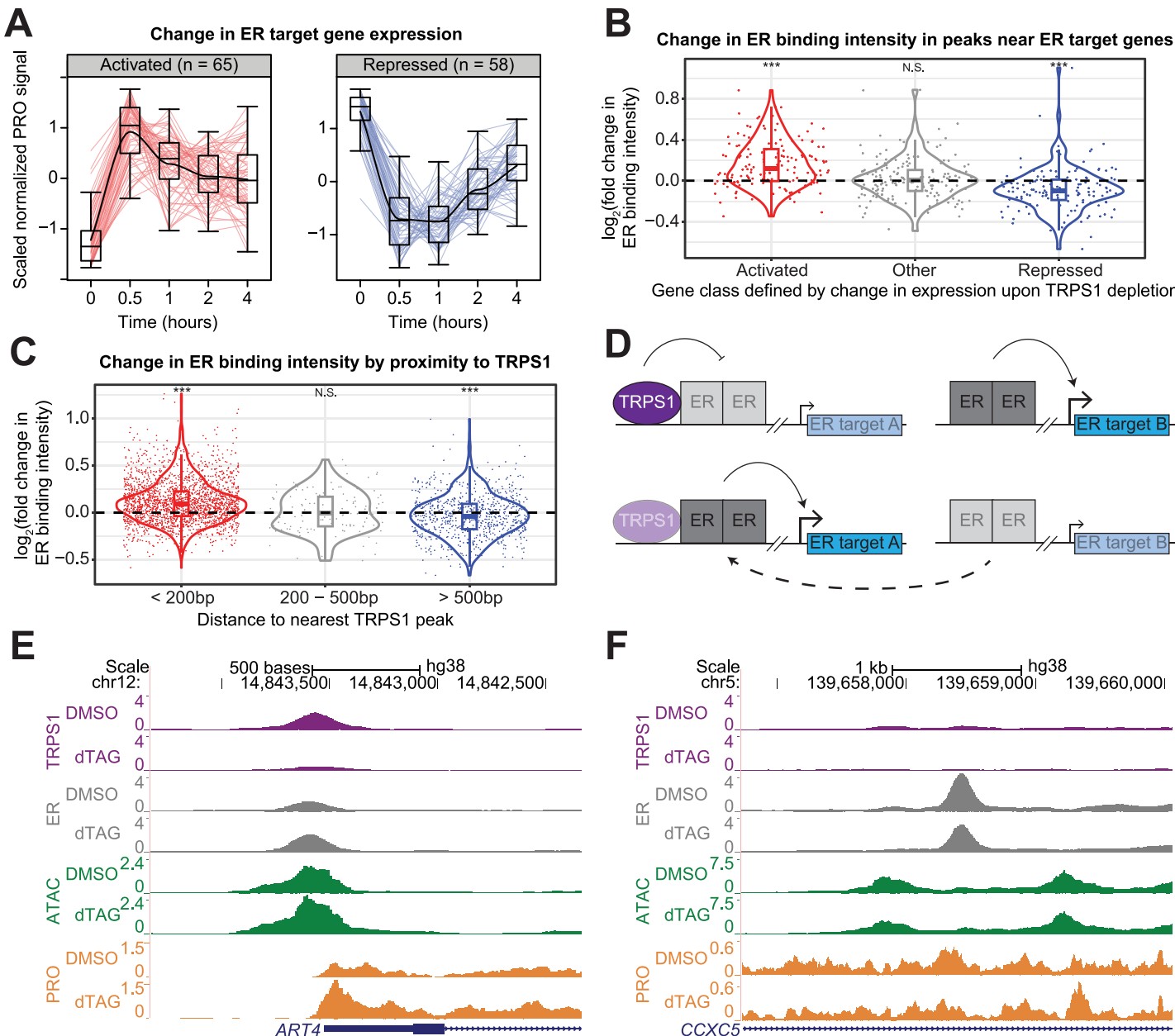

**Fig 5. TRPS1 redistributes ER binding to modulate ER target gene transcription.** A: Kinetic traces of activated and repressed ER target genes over the time course. B: Violin and box and whisker plots for ER binding intensity fold change upon TRPS1 depletion at ER ChIP-seq peaks within 100kb of the TSS of each gene, grouped by gene class defined by change in expression upon TRPS1 depletion. (In (B) and (C), *** represents a significant one-sample t-test p-value $< 10^{-3}$. N.S. represents a non-significant p-value $> 0.1$.) C: Violin and box and whisker plots for ER binding intensity fold change upon TRPS1 depletion at ER ChIP-seq peaks, grouped by summit-to-summit distance to the nearest TRPS1 ChIP-seq peak. D: Model of TRPS1-mediated ER redistribution and modulation of ER target gene transcription. Transparent boxes indicate reduced binding intensity or attenuated transcription. Above, at baseline, TRPS1 directly decreases ER binding intensity proximal to TRPS1, attenuating ER activation of proximal ER target genes. Distal to TRPS1, ER binding intensity is not directly affected by TRPS1, and ER fully activates proximal ER target genes. Below, after TRPS1 depletion, ER binding proximal to TRPS1 increases in intensity, augmenting ER activation of proximal ER target genes. Distal to TRPS1, ER binding intensity is indirectly decreased, as limiting ER molecules are redistributed to TRPS1-proximal regulatory elements, attenuating ER activation of proximal ER target genes. E: ChIP, ATAC, and PRO density around an example increased ER binding site near an activated ER target gene. At this TRPS1-proximal ER binding site, upon dTAG treatment, TRPS1 binding intensity decreases, ER binding intensity increases, chromatin accessibility increases, and gene expression increases. F: ChIP, ATAC, and PRO density around an example decreased ER binding site near an repressed ER target gene. At this TRPS1-distal ER binding site, upon dTAG treatment, ER binding intensity decreases, chromatin accessibility decreases, and gene expression decreases. In (F) and (G), dTAG refers to dTAG-13 and dTAG$^V$-1 at 50nM each.

binding from TRPS1-distal sites to TRPS1-proximal sites and modulates ER target gene transcription proximal to the dynamic ER binding sites (Fig 5D). To illustrate this phenomenon, we provide an example of a TRPS1-proximal, increased ER peak near an activated ER target gene in (Fig 5E), and a TRPS1-distal, decreased ER peak near a repressed ER target gene in (Fig 5F).

### TRPS1 activity is associated with breast cancer patient outcomes

We next sought to connect the primary TRPS1-responsive genes with downstream cellular and patient-related outcomes. We defined a new steady state of transcription with the 24 hour time point after TRPS1 depletion. We ranked genes based on their shrunken fold change in PRO signal (Fig 6A). Using this ranking, we performed gene set enrichment analysis with the Hallmark gene sets and found multiple cell-cycle-related gene sets to be negatively enriched, including E2F Targets (Fig 6B) [44, 45]. Consistent with this, we observed a significant decrease in cell number doubling rate of T47D dTAG-TRPS1 cells upon TRPS1 depletion (Fig 6C). Importantly, the isogenic parental T47D cells do not display a cell number defect with dTAG treatment, so we attribute this effect to TRPS1 depletion and not a non-specific effect of the compounds (S10 Fig).

Finally, we calculated a TRPS1 activity score by adapting methods developed by [48, 49]. We used our PRO-seq data to determine a primary TRPS1 regulon based on the differentially expressed genes 30 minutes after TRPS1 depletion. We classified breast cancer patients from the METABRIC cohort as having high TRPS1 activity if both a) TRPS1-repressed genes are negatively enriched and b) TRPS1-activated genes are positively enriched, relative to all other patients in the cohort (example patient in (Fig 6D)) [46, 47]. Similarly, we classified patients as having low TRPS1 activity if both a) TRPS1-repressed genes are positively enriched and b) TRPS1-activated genes are negatively enriched. We classified the remaining patients as having intermediate TRPS1 activity. We ranked patients based on their TRPS1 activity and found no association with other clinical covariates (Fig 6E). When we stratified patients by TRPS1 activity, we found TRPS1 activity to be significantly associated with shorter survival time (Logrank p-value $4.99*10^{-4}$) (Fig 6F). When we first separated tumors by ER-positivity or by intrinsic subtype, we found that this association was specific for ER-positive tumors and Luminal A tumors (S11 Fig). We also performed this analysis using genes differentially expressed after 24 hours of TRPS1 depletion (S12 Fig). However, by this time cell cycle genes dominate, and we speculate that the association with breast cancer patient survival is due to these genes. We believe using the primary response genes offers a unique measure of TRPS1 activity not achievable using previous surrogates.

## Discussion

In this study, we used rapidly inducible targeted protein degradation to systematically determine the primary effects of acute TRPS1 depletion on chromatin accessibility, ER binding, and nascent transcription in a luminal breast cancer cell line. We focused on TRPS1 based on two orthogonal, genome-wide, unbiased assays that implicated *TRPS1* in the processes of breast tumor incidence and breast cancer cell number accumulation.

First, we used the summary statistics from a recent GWAS to plot two sets of common genetic variants in the *TRPS1* locus associated with breast cancer incidence [28]. These genetic variants were independently identified as significantly associated with breast cancer incidence in a previous GWAS [50], but Zhang *et al.* determined that the association was strongest among luminal breast tumors. Second, we analyzed data from the Cancer Dependency Map project and found that sensitivity to *TRPS1* knockout was correlated with sensitivity to *ESR1*

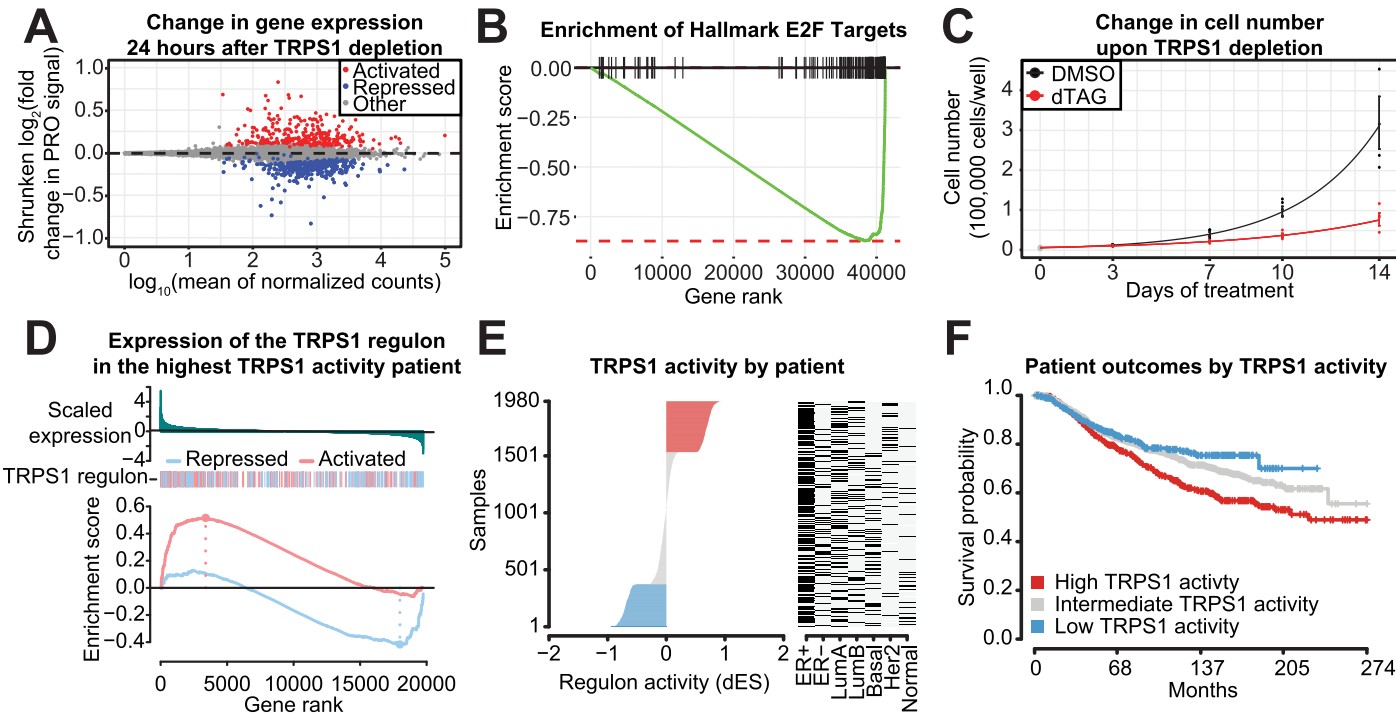

**Fig 6. TRPS1 activity is associated with breast cancer patient outcomes.** A: MA plot of PRO signal in each gene, with shrunken log fold change values representing transcription in the 30 minute dTAG-13 and dTAG$^V$-1 at 50nM each (dTAG) treatment condition relative to the DMSO condition. B: Mountain plot of the Hallmark E2F Targets gene set, using genes ranked by shrunken fold change from (A). A negative enrichment score indicates an enrichment of the gene set among repressed genes. Adjusted p-value $2.5*10^{-19}$. C: Cell number over time of dTAG-TRPS1 cells treated with dTAG or DMSO. Analysis of variance for the coefficient corresponding to the difference in doubling rates between the conditions in a linear model of the logarithm of cell number versus time: p-value $1.1*10^{-5}$. D: Differential enrichment score (dES) calculation for an example patient with the highest TRPS1 activity. Above, genes ranked by scaled expression in this patient relative to all other patients in the METABRIC cohort [46, 47]. Below, gene set enrichment analysis of TRPS1-repressed and TRPS1-activated genes defined by response after 30 minutes of TRPS1 depletion. E: Patients from the METABRIC cohort, ranked by dES as calculated in (D), with classifications of the tumors on the right. F: Kaplan-Meier curves for patients in the METABRIC cohort, stratified by TRPS1 activity as in (E). Logrank p-value $4.99*10^{-4}$.

knockout and significantly enriched among luminal breast cancer cell lines [31]. Both of these unbiased screens indicate that *TRPS1* contributes to luminal breast cancer cell fitness and led us to the hypothesis that TRPS1 influences ER activity or genomic binding.

As TFs regulate the transcription of many other chromatin-associated factors that themselves regulate RE activity, TF binding, and transcription, we sought to isolate the primary effects of TRPS1 depletion. To do so, we used the dTAG inducible degron tag system to acutely deplete endogenous TRPS1 protein abundance within minutes of induction [32]. This is in contrast to traditional RNA interference or gene knockout methods, which can take days to deplete the target of interest.

Minutes to hours after TRPS1 depletion, we performed several sensitive, genome-wide assays. ATAC-seq and ChIP-seq can be performed at any time point after a perturbation, as they measure chromatin accessibility and chromatin-associated factor binding, which can change with rapid kinetics. In contrast, changes in messenger RNA abundance accumulate more slowly, with kinetics that depend not only on the rate of nascent transcription but also on the ratio of abundance to synthesis and degradation rates. In contrast, nascent transcriptional profiling measures the immediate change in RNA synthesis rates after a perturbation. Here we use PRO-seq coupled with acute TRPS1 depletion to identify primary TRPS1-responsive genes.

With our cell lines and assays in hand, we first measured changes in chromatin accessibility upon TRPS1 depletion. Consistent with previous studies linking TRPS1 to corepressor complexes, we found that the predominant effect of TRPS1 depletion is an increase in chromatin accessibility and bidirectional transcription at REs [14–18]. We observed decreasing enrichment of the GATA motif prevalence in increased peaks over time, indicating that our shortest time point was the most specific for isolating the primary effects of TRPS1 depletion. Intriguingly, we identified a significant enrichment of ER half-site motifs in increased as well as decreased ATAC peaks, suggesting that ER binding intensity was changing in a site-specific manner.

We next measured changes in nascent transcription minutes to hours after TRPS1 depletion and clustered the gene responses. Activated genes were enriched for cholesterol homeostasis genes. Of note, several recent GWAS have identified SNPs in the *TRPS1* locus associated with blood cholesterol levels [51–53]. However, as of yet, no mechanistic follow-up studies into the role of TRPS1 in cholesterol biology have been performed. After TRPS1 depletion, repressed genes were enriched for estrogen response gene sets. Consistent with our previous data, activated genes were closer to increased TRPS1-bound ATAC peaks, suggesting TRPS1 directly represses these target genes at steady state. On the other hand, repressed genes were not closer to decreased TRPS1-bound ATAC peaks, suggesting a distinct mechanism of transcriptional regulation of this gene class.

We hypothesized that at steady state TRPS1 directly represses and indirectly activates its primary response genes. Using ER target genes and ER genomic binding as a case study, we found evidence supporting a model of acute ER redistribution. ER binding sites proximal to TRPS1 tended to increase in intensity upon TRPS1 knockdown, with distal ER binding sites tending to decrease in intensity. Furthermore, genes activated upon TRPS1 depletion were surrounded by ER binding sites that increased in intensity, and repressed genes were near decreased ER binding sites. Taken together, we propose a model in which TRPS1 directly decreases chromatin accessibility at steady state. Upon acute TRPS1 depletion, TRPS1-proximal REs increase in accessibility, an effect which we propose allows ER to redistribute from TRPS1-distal REs. In this proposed model, subsets of ER target genes are activated or repressed by TRPS1 depletion via distinct mechanisms.

First described in the 1980s, the concept of coactivator "squelching" has been debated as a mechanism of indirect activity distal from a TF's genomic binding sites [54, 55]. Squelching has been proposed as a mechanism by which nuclear receptors like ER acutely repress transcription of a subset of primary response genes by competing for limiting coactivators [43, 56–58]. Here we propose not the redistribution of coactivators by an activating TF, but a redistribution of activating TFs themselves via a rapid increase in local chromatin accessibility after the acute depletion of a repressive TF.

Our findings of both increased and decreased ER genomic binding and target gene transcription are distinct from previous studies of the effects of TRPS1 on TF binding and activity. Elster *et al.* used an unbiased screen to identify TRPS1 as a repressor of YAP1 activity in another luminal breast cancer cell line, MCF7 [17]. After TRPS1 knockdown, the authors observed a genome-wide activation of YAP target genes. We did not find a YAP gene signature among dynamic genes in our PRO-seq data, though we did observe an enrichment of TEAD motifs in increased ATAC peaks, suggesting differences between the cells used in each study and perhaps their baseline YAP-TEAD activity. While we would predict that our acute redistribution model is generalizable to other TFs and sets of TRPS1-regulated genes beyond ER and its target genes, it remains possible that TRPS1 modulates the genomic binding intensity of additional TFs in a unidirectional manner via a distinct mechanism.

Serandour *et al.* knocked down *TRPS1* in MCF7 cells and reported both a genome-wide repression of ER target genes as well as a genome-wide increase in ER binding [16]. We did not perform ChIP-seq at a comparable time point to their days-long knockdown, so we cannot directly compare our ER binding data. Our latest PRO-seq time point was 24 hours after TRPS1 depletion, at which time we do not observe a genome-wide repression of ER target genes. This could once again be attributable to a difference in cell lines. However, we would also speculate that the unidirectional and nonconcordant changes in ER binding and target gene expression at later time points could be due to non-primary effects of extended *TRPS1* knockdown.

Finally, we used PRO-seq data from both late and early time points to identify genes that represent cells at a new steady state after TRPS1 depletion, as well as primary TRPS1-responsive genes. After 24 hours of TRPS1 depletion, repressed genes were enriched for cell cycle related genes, consistent with a decrease in cell number doubling rate. Unique to this study, we used primary TRPS1-responsive genes to define a TRPS1 activity score, adapting a method based on predicted TF target genes [48, 49]. Using this method, we were able to stratify breast cancer patients into groups with differing survival probabilities.

Using TRPS1 activity score to classify patients may provide additional insight into the transcriptional program within a patient's tumor that might not be immediately apparent based on previous surrogates for TRPS1 activity. For example, *TRPS1* is frequently amplified in breast tumors, and this amplification is associated with worse prognosis [16, 59]. However, *TRPS1* is often co-amplified along with the rest of the chromosomal segment 8q23–q24, where the proto-oncogene *MYC* resides, making it difficult to discern whether *TRPS1* amplification is a driver of breast cancer progression [60].

In contrast, higher *TRPS1* expression has been associated with better breast cancer patient outcomes, though its expression is highly correlated with ER and GATA3, both favorable prognostic indicators [22, 61]. As relative TF expression and activity across patients are not always identical, our data uses primary TRPS1-responsive genes as a measure of TRPS1 activity. Our TRPS1 activity score is not correlated with ER-positivity and effectively stratifies patients. Though our patient outcome analysis, as with all similar analyses, describes an association and does not necessarily imply a causative relationship, the direction is consistent with the effect on cell number observed in this study as well as the Cancer Dependency Map, suggesting that TRPS1 drives breast cancer cell number accumulation.

Altogether, we provide a systematic study of the primary effects of rapid TRPS1 depletion in luminal breast cancer cells. We propose a model in which TRPS1 depletion leads to decondensation of local chromatin structure, allowing for the acute redistribution of ER, both activating and repressing subsets of ER target genes. This TRPS1-regulated transcription appears to be relevant for cancer cell fitness, as TRPS1 depletion decreases cell number doubling rate, and high TRPS1 activity is associated with worse breast cancer patient outcomes. These methods of inducible targeted protein degradation coupled with genomic chromatin assays and nascent RNA transcriptional profiling should in principle be applicable to the study of any TF, allowing us to better understand the mechanisms behind the phenotypes associated with additional GWAS hits.

## Materials and methods

### GWAS and DepMap data visualization

Summary statistics from [28] were downloaded from the NHGRI-EBI GWAS Catalog [29]. SNPs in the *TRPS1* locus were plotted using LocusZoom [30]. Knockout scores and luminal

breast cancer identifiers were downloaded from the Cancer Dependency Map project [31] and plotted using the statistical programming language R [62].

## Cell culture

T47D cells (RRID:CVCL_0553) (ATCC) were cultured in RPMI 1640 medium (Gibco) supplemented with 10% fetal bovine serum (Gemini) and 10µg/ml insulin from bovine pancreas (Sigma, made as a 1000x solution in 1% aqueous glacial acetic acid).

## Plasmid generation for gene editing

DNA for transfection was prepared as previously described [63, 64]. A CRISPR sgRNA (TTATCTTTGCAGATATGGTC) targeting the 5' end of the *TRPS1* coding sequences was designed using Benchling. The sgRNA was cloned into hSpCas9 plasmid PX458 (Addgene #48138) as previously described [65], using the following primers:

5' -CACCGTTATCTTTGCAGATATGGTC-3' and
    5' -AAACGACCATATCTGCAAAGATAAC-3'.

A plasmid harboring a synthetic HygR-P2A-2xHA-FKBP_F36V insert was generated with Cold Fusion (System Biosciences), starting with the HygR-P2A-AID cassette in pMGS58 (Addgene #135311) [64] and the Puro-P2A-2xHA-FKBP_F36V casette in (Addgene #91793) [32]. The linear donor was generated by PCR using primers (IDT) that contain 50-nucleotide homology tails and gel-purified. The primers contained 5' phosphorothioate modifications to increase PCR product stability in the cell [66]. The primers used for making PCR donor fragments were:

5' -G*T*AACTTTCAGATAACACTGTATCTGCCTTTT
    CCCTTTATCTTTGCAGATATGAAAAAGCCTGAACT
    CACCG-3'
    and
    5' -T*T*CACTTGCAACGTTTCTCAGAGGGGGGTTCT
    TTTTCCGGACACCTGAACCTGAACCTCCAGATCCA
    CCAGATCTTTCCAGTTTTAGAAGCTCCACATCG-3'
    with asterisks representing the phosphorothioate modifications.

## dTAG-TRPS1 clone generation

Clones were generated as previously described [63, 64], with modifications. An initial round of cloning was performed using puromycin selection, but upon genomic DNA sequencing this clone did not appear to have a dTAG insertion event within *TRPS1*. Nevertheless, this clone was used for a second round of cloning using hygromycin selection. $3*10^6$ cells were plated in 10cm plates. The next day, cells were cotransfected with 15µg of CRISPR/Cas9-sgRNA plasmid and 1.85µg of linear donor PCR product using Lipofectamine 3000 (Thermo Fisher Scientific) in Optimem (Gibco). One day after transfection, the media was replaced. Starting four days after transfection, cells were selected for two weeks with 200µg/mL of Hygromycin B (Invitrogen) with 20% conditioned media, replaced twice per week. Colonies were then grown in 20% conditioned media, replaced twice per week, until they were large enough to be picked and passaged to a 24-well plate. Clones were expanded and frozen at 8 passages after transfection. Integration was tested with Western blotting, PCR, and Sanger sequencing. In each of the three clones, two to three of the four genomic copies of *TRPS1* are knocked out, and only

tagged TRPS1 protein is expressed. Details about each of the determined alleles are available at
https://guertinlab.github.io/TRPS1_ER_analysis/Vignette.html#allele-sequencing.

## Western blotting

$8*10^5$ cells per sample were plated in each well of a 6-well plate. Cells were treated with DMSO
or 50nM dTAG-13 and 50nM dTAG$^V$-1 in DMSO at various time points and collected simul-
taneously. At the time of harvest, cells were scraped and lysed in RIPA buffer (1% Nonidet P-
40, 1% sodium deoxycholate, 0.1% sodium dodecyl sulfate, 2mM EDTA, 150mM NaCl, 10mM
sodium phosphate, 50mM NaF, 50mM Tris pH 7.5), with 100µM benzamidine, 5µg/mL apro-
tinin, 5µg/mL leupeptin, 1µg/mL pepstatin, 1mM phenylmethylsulfonyl fluoride, and 2mM
sodium orthovanadate added fresh. Lysates were sonicated in a Biorupter UCD-200 (Diage-
node) on high for 30 seconds on and 30 seconds off for 5 cycles, and clarified by centrifugation
at 14,000rpm for 15 min in 4˚C. Protein concentration was measured by BCA assay and
diluted to the same concentration. 10× Laemmli buffer was added to a final concentration of
1x, and 2-mercaptoethanol was added to a final concentration of 1%. Samples were boiled at
95˚C for 10 minutes, and 30µg of each was loaded into a 10% polyacrylamide gel. Samples
were separated by gel electrophoresis and transferred to nitrocellulose membranes. Mem-
branes were incubated in blocking buffer (3% bovine serum albumin, 1X Tris buffered saline)
for 1 hour at room temperature with rocking. Primary antibodies (anti-TRPS1, Cell Signaling
#17936S, and anti-ACTB, Cell Signaling #3700S) were diluted 1:1,000 in primary buffer (3%
bovine serum albumin, 0.1% sodium azide, 0.1% Tween-20, 1X Tris buffered saline) at 4˚C
with rocking overnight. Fluorescent secondary antibodies were diluted 1:10,000 in secondary
buffer (5% bovine serum albumin, 0.1% sodium azide, 0.1% Tween-20, 1X Tris buffered saline)
and incubated for 1 hour at room temperature with rocking, and fluorescence was measured
(Odyssey, Licor).

## ChIP-seq library preparation

$2.4*10^7$ cells per sample were plated across 3 15cm dishes 2 days before harvest. Cells were
treated with DMSO or 50nM dTAG-13 and 50nM dTAG$^V$-1 in DMSO for 30 minutes and col-
lected simultaneously. At the time of harvest, cells were fixed with 1% formaldehyde (Sigma)
for 10 minutes at 37˚C and quenched with 125mM Glycine (Fisher) for 10 minutes at 37˚C.
Plates were moved to ice, and cells were washed and scraped into ice cold PBS containing
Complete EDTA-free Protease Inhibitor Cocktail (Roche). Cells were pelleted in aliquots of
$3.6*10^7$ cells, snap frozen in liquid nitrogen, and stored at -80˚C. Pellets were thawed, and cells
were lysed in 1mL Cell Lysis Buffer (85mM KCl, 0.5%NP40, 5mM PIPES pH 8.0), with prote-
ase inhibitor cocktail added fresh, for 10 minutes with rotation at 4˚C. Nuclei were pelleted at
3300g at 4˚C for 5 minutes and resuspended in 500µL ChIP lysis buffer (0.5% SDS, 10mM
EDTA, 50mM Tris-HCl pH 8.1), with protease inhibitor cocktail added fresh, for 10 minutes
with rotation at 4˚C. Lysates were moved to 15ml polystyrene conical tubes (Falcon) and soni-
cated in a Biorupter UCD-200 (Diagenode) on high for 30 seconds on and 30 seconds off for 4
sets of 5 cycles. Before each set, ice in the water bath was replaced, and samples were gently
vortexed to mix. Sonicated lysates were then move to 1.5ml tubes and clarified by centrifuga-
tion at 14,000rpm for 15 min in 4˚C. 500µL of the supernatant was diluted into 6.5mL Dilution
Buffer (0.01% SDS, 1.1% Triton X-100, 1.2mM EDTA, 167mM NaCl, 16.6mM Tris-HCl pH
8.0), with protease inhibitor cocktail added fresh ($1*10^6$ cells in 200µL). 1ml ($5*10^6$ cells) was
aliquoted into each of 3 tubes with antibody (1.25 µg anti-HA, Cell Signaling #3724S, 2.5µg
anti-ER, Millipore #06–935, or 2.5µg IgG control, Cell Signaling #2729S), and incubated with
end-over-end rotation at 4˚C overnight.

50μL Protein A/G Magnetic Beads (Pierce) per sample were washed with bead washing buffer (PBS with 0.1% BSA and 2mM EDTA) and then incubated with samples for 2 hours with rotation at 4°C. The samples were washed once each with low salt immune complex buffer (0.1% SDS, 1% Triton x-100, 2mM EDTA, 150mM NaCl, 20mM Tris HCl pH 8.0), high salt immune complex buffer (0.1% SDS, 1% Triton x-100, 2mM EDTA, 500mM NaCl, 20mM Tris Hcl pH8.0), LiCl immune complex buffer (0.25M LiCl, 1% NP-40, 1% deoxycholate, 1mM EDTA, 10mM Tris-HCl pH8.0), and 1xTE (10mM Tris-HCl, 1mM EDTA pH8.0). Immune complexes were eluted in elution solution, (1% SDS, 0.1M sodium bicarbonate) in a thermomixer for 30 min at 65°C at 1,200rpm. Crosslinks were reversed and proteins were digested with the addition of 200mM NaCl and 2ul Proteinase K in a thermocycler at 65°C for 16 hours. DNA was purified with a Qiaquick PCR cleanup (Qiagen), and libraries were prepared with a NEBNext Ultra II Library Prep Kit (New England Biolabs).

## ChIP-seq analysis

Adapters were removed using `cutadapt` [67]. Reads were aligned to the *hg38* genome assembly with `bowtie2` [68]. Duplicate reads were removed, and the remaining reads were sorted into *BAM* files and converted to *bed* format for counting with `samtools` [69]. Reads were also converted to *bigWig* format with `deeptools` [70]. Peaks were called with `MACS2` [35]. Reads were counted in peaks using `bedtools`, and differentially bound peaks were identified with `DESeq2` [37, 71]. Heatmaps were generated with `deeptools`. Peak proximity to and overlap with other features were calculated with `bedtools`.

## ATAC-seq library preparation

ATAC-seq libraries were prepared as previously described [72], with modifications. 4 replicates were performed from cells treated and collected at different times in the same day. $4*10^5$ cells per sample were plated in each well of a 6-well plate 2 days before harvest. Cells were treated with DMSO or 50nM dTAG-13 and 50nM dTAG$^V$-1 in DMSO at various time points and collected simultaneously. At the time of harvest, cells were moved to ice and scraped in 1mL ice cold PBS, and 100μL ($\sim 5 \times 10^4$ cells) were transferred to 1.5 mL tubes. Cells were centrifuged at 500 x $g$ for 5 minutes at 4°C, and the pellets were resuspended in 50 μL cold lysis buffer (10mM Tris-HCl, 10mM NaCl, 3mM MgCl$_2$, 0.1% NP-40, 0.1% Tween-20, 0.01% Digitonin, adjusted to pH 7.4) and incubated on ice for 3 minutes. Samples were washed with 1 mL cold wash buffer (10mM Tris-HCl, 10mM NaCl, 3mM MgCl$_2$, 0.1% Tween-20). Cells were centrifuged at 500 x $g$ for 10 minutes at 4°C, and pellets were resuspended in the transposition reaction mix (25 μL 2X TD buffer (Illumina), 2.5 μL TDE1 Tn5 transposase (Illumina), 16.5 μL PBS, 0.5 μL 1% Digitonin, 0.5 μL 10% Tween-20, 5 μL nuclease-free water) and incubated in a thermomixer at 37°C and 100rpm for 30 minutes. DNA was extracted with the DNA Clean and Concentrator-5 Kit (Zymo Research). Sequencing adapters were attached to the transposed DNA fragments using NEBNext Ultra II Q5 PCR mix (New England Biolabs), and libraries were amplified with 8 cycles of PCR. PEG-mediated size fractionation [73] was performed on the libraries by mixing SPRIselect beads (Beckman) with each sample at a 0.5:1 ratio, then placing the reaction vessels on a magnetic stand. The right side selected sample was transferred to a new reaction vessel, and more beads were added for a final ratio of 1.8:1. The final size-selected sample was eluted into nuclease-free water. This size selection protocol was repeated to further remove large fragments.

## ATAC-seq analysis

Adapters were removed using `cutadapt` [67]. Reads aligning to the mitochondrial genome with `bowtie2` [68] were removed. The remaining reads were aligned to the *hg38* genome assembly with `bowtie2`. Duplicate reads were removed, and the remaining reads were sorted into *BAM* files with `samtools` [69]. Reads were converted to *bed* format with `seqOutBias` and *bigWig* format with `deeptools` [70, 74]. Accessibility peaks were called with `MACS2` [35]. Reads were counted in peaks using `bedtools`, and differentially accessible peaks were identified with `DESeq2` [37, 71]. *de novo* motif identification was performed on dynamic peaks with `MEME`, and `TOMTOM` was used to match motifs to the HOMER, Jaspar, and Uniprobe TF binding motif databases [75–78]. `AME` was used to identify motifs enriched in increased or decreased peaks relative to unchanged peaks [79]. `FIMO` and `bedtools` were used to assess motif enrichment around peak summits [80]. Dynamic peaks were clustered into response groups using `DEGreport` [81].

## PRO-seq library preparation

Cell permeabilization was performed as previously described [82], with modifications. 4 replicates were performed from cells treated and collected at different times in the same day. For the time course experiment, $8*10^6$ dTAG-TRPS1 Clone 28 cells per sample were plated in 15cm dishes 2 days before harvest. Cells were treated with DMSO or 50nM dTAG-13 and 50nM dTAG$^V$-1 in DMSO at various time points and collected simultaneously. For the three clone experiment, $4*10^6$ cells per sample were plated in 10cm dishes 1 day before harvest. Cells were treated with DMSO or 100nM dTAG-13 in DMSO for 90 minutes and collected simultaneously.

At the time of harvest, cells were scraped in 10mL ice cold PBS and washed in 5mL buffer W (10mM Tris-HCl pH 7.5, 10mM KCl, 150mM sucrose, 5mM $MgCl_2$, 0.5mM $CaCl_2$, 0.5mM DTT, 0.004U/mL SUPERaseIN RNase inhibitor (Invitrogen), Complete protease inhibitors (Roche)). Cells were permeabilized by incubating with buffer P (10 mM Tris-HCl pH 7.5, KCl 10 mM, 250 mM sucrose, 5 mM $MgCl_2$, 1 mM EGTA, 0.05% Tween-20, 0.1% NP40, 0.5 mM DTT, 0.004 units/mL SUPERaseIN RNase inhibitor (Invitrogen), Complete protease inhibitors (Roche)) for 3 minutes on ice. Cells were washed with 10 mL buffer W before being transferred into 1.5mL tubes using wide bore pipette tips. Finally, cells were resuspended in 50μL buffer F (50mM Tris-HCl pH 8, 5mM $MgCl_2$, 0.1mM EDTA, 50% Glycerol, 0.5 mM DTT). Cells were snap frozen in liquid nitrogen and stored at -80˚C.

PRO-seq libraries were prepared as previously described [83], with modifications. RNA extraction after the run-on reaction was performed with 500μL Trizol LS (Thermo Fisher) followed by 130μL chloroform (Sigma). The equivalent of 1μL of 50μM for each adapter was used. A random eight base unique molecular identifier (UMI) was included at the 5' end of the adapter ligated to the 3' end of the nascent RNA. 37˚C incubations were performed with rotation with 1.5mL tubes placed in 50mL conical tubes in a hybridization oven. For the reverse transcription reaction, RP1 was used at 100μM and dNTP mix was used at 10mM each. Libraries were amplified by PCR for a total of 8 cycles in 100μL reactions with Phusion polymerase (New England Biolabs). No PAGE purification was performed to ensure that our libraries were not biased against short nascent RNA insertions.

## PRO-seq analysis

Adapters were removed using `cutadapt` [67]. Libraries were deduplicated using `fqdedup` and the 3' UMIs [84]. UMIs were removed, and reads were reverse complemented with the `seqtk`. Reads aligning to the rDNA genome with `bowtie2` [68] were removed. The

remaining reads were aligned, sorted, and convert to *bed* and *bigWig* files with `bowtie2`, `samtools`, `seqOutBias`, and `deeptools`, respectively [69, 70, 74]. Composite profiles around TRPS1 peaks were generated with `deeptools`. Reads were counted in genes using `bedtools`, and differentially expressed genes were identified with `DESeq2` [37, 71]. Dynamic genes were clustered into response groups using `DEGreport` [81]. Over-representation analysis was performed with `enrichr` [85], and gene set enrichment analysis was performed with `fgsea` [86], both using the Hallmark gene sets [41].

## Genome browser visualization

Genome browser [87] images were taken from the following session:https://genome.ucsc.edu/s/tgscott/dTAG_TRPS1_ChIP_PRO_ATAC.

## Cell number enumeration

$1.25*10^4$ cells per sample were plated in a 24-well plate. The next day (day 0), cells were treated with DMSO or 50nM dTAG-13 and 50nM dTAG$^V$-1 in DMSO. Media was replaced, maintaining the treatment condition, every 2 days. Cells were enumerated using a hemocytometer. 2 technical replicates were used on day 0, 2 for each treatment on day 3, and 3 for each treatment on days 7, 10, 14. The technical replicates were merged, and the experiment was performed in 4 biological replicates from different cell passages. The data were imported into R [62] for visualization and statistical analysis. A linear model was fit for the log-transformed cell number and the time. A second linear model was fit that included an interaction term between the time and the treatment condition, representing the effect of treatment on the doubling rate. Analysis of variance was performed on the two models to test for the significance of the interaction term.

## TRPS1 activity score and patient outcome stratification

Primary TRPS1-regulated genes were defined based on the 30 minute time point using `DESeq2` [37]. This TRPS1 regulon was then used in `RTN` [48, 49] to define a TRPS1 activity score for each patient within the METABRIC cohort [46, 47].

## Supporting information

**S1 Fig. Fraction of reads in peaks (FRiP) for ChIP-seq libraries.** FRiP scores for each library, calculated using the ChIPQC R package [34].
(PDF)

**S2 Fig. Strand cross-correlation (CC) plots for ChIP-seq libraries.** CC values for each library, calculated using the ChIPQC R package [34].
(PDF)

**S3 Fig. Fragment size distribution plots for ATAC-seq libraries.** A plot for each library was generated using the ATACseqQC R package [36].
(PDF)

**S4 Fig. Plots of signal enrichment around TSS's for ATAC-seq libraries.** A plot for each library was generated using the ATACseqQC R package [36].
(PDF)

**S5 Fig. Quality control metrics for PRO-seq libraries.** Quality control metrics are defined as in [39]. Each metric is a row, and each sample is a column. The green region for each metric is

the goal for a high quality library.
(PDF)

**S6 Fig. Bidirectional transcription at TRPS1 peaks increases upon TRPS1 depletion.** MA plot of TRPS1 ChIP-seq peaks from Fig 3G, with fold change values representing bidirectional transcription in the 30 minute dTAG-13 and dTAGV -1 at 50nM each (dTAG) treatment condition relative to the DMSO condition. Testing for a TRPS1 cistrome-wide increase in bidirectional transcription, the ANOVA F-test p-value was $< 2.2*10^{-16}$.
(PDF)

**S7 Fig. Chromatin accessibility at ATAC-seq peaks without bidirectional transcription increases upon TRPS1 depletion.** MA plot of ATAC-seq peaks, with fold change values representing accessibility in the 30 minute dTAG-13 and dTAGV -1 at 50nM each (dTAG) treatment condition relative to the DMSO condition, as in Fig 3.
(PDF)

**S8 Fig. Acute estrogen treatment identifies direct ER target genes in T47D cells.** MA plot of PRO signal, with fold change values representing transcription in the 90 minute estrogen treatment condition relative to the DMSO condition. Each point represents a gene, and black points represent the estrogen-activated genes that we use in Fig 5.
(PDF)

**S9 Fig. The effect of TRPS1 depletion on ER target gene transcription is consistent across three independent clones.** Fold changes in normalized PRO signal across three independent clones for ER target genes that are (A) activated or (B) repressed upon TRPS1 depletion, as defined in Fig 5A.
(PDF)

**S10 Fig. T47D cells do not display a cell number defect with dTAG treatment.** Cell number over time of parental T47D cells treated with dTAG or DMSO, as in Fig 6C.
(PDF)

**S11 Fig. TRPS1 activity is associated with breast cancer patient outcomes specifically for ER-positive and Luminal A tumors.** A) Kaplan-Meier curves for patients in the METABRIC cohort, stratified by TRPS1 activity as in Fig 6F, separated by ER-posivity. Logrank p-value $3.83*10^{-6}$ for ER-positive tumors and not significant for ER-negative tumors. B) Kaplan-Meier curves for patients in the METABRIC cohort, stratified by TRPS1 activity as in Fig 6F, separated by intrinsic subtype. Logrank p-value $4.23*10^{-4}$ for Luminal A tumors and not significant for the other subtypes.
(PDF)

**S12 Fig. Genes differentially expressed after 24 hours of TRPS1 depletion are associated with breast cancer patient outcomes.** Kaplan-Meier curves for patients in the METABRIC cohort, stratified by TRPS1 activity as in Fig 6F, but using genes differentially expressed after 24 hours of TRPS1 depletion. Logrank p-value $2.09*10^{-13}$.
(PDF)

**S1 Table. ATAC-seq peaks that increase in accessibility upon 30 minutes of TRPS1 depletion (red dots in Fig 3A).**
(TXT)

**S2 Table. ATAC-seq peaks that decrease in accessibility upon 30 minutes of TRPS1 depletion (blue dots in Fig 3A).**
(TXT)

**S3 Table. Top 50 motifs significantly enriched in increased ATAC peaks relative to unchanged ATAC peaks at 30 minutes.** Results generated using AME [79].
(CSV)

**S4 Table. Motifs significantly enriched in decreased ATAC peaks relative to unchanged ATAC peaks at 30 minutes.** Results generated using AME [79].
(CSV)

**S5 Table. Genes that are gradually activated upon TRPS1 depletion (from Fig 4A).**
(TXT)

**S6 Table. Genes that are transiently activated upon TRPS1 depletion (from Fig 4A).**
(TXT)

**S7 Table. Genes that are gradually repressed upon TRPS1 depletion (from Fig 4B).**
(TXT)

**S8 Table. Genes that are transiently repressed upon TRPS1 depletion (from Fig 4B).**
(TXT)

**S9 Table. ER target genes that are activated upon TRPS1 depletion (from Fig 5A).**
(TXT)

**S10 Table. ER target genes that are repressed upon TRPS1 depletion (from Fig 5A).**
(TXT)

## Acknowledgments

We thank all members of the Guertin and Gioeli laboratories, especially Arun Dutta, Jacob Wolpe, Devin Roller, and Adam Spencer, for critical feedback.

## Author Contributions

**Conceptualization:** Thomas G. Scott, Kizhakke Mattada Sathyan, Daniel Gioeli, Michael J. Guertin.

**Data curation:** Thomas G. Scott.

**Formal analysis:** Thomas G. Scott, Daniel Gioeli, Michael J. Guertin.

**Funding acquisition:** Michael J. Guertin.

**Investigation:** Thomas G. Scott.

**Methodology:** Thomas G. Scott, Kizhakke Mattada Sathyan, Daniel Gioeli, Michael J. Guertin.

**Project administration:** Michael J. Guertin.

**Resources:** Michael J. Guertin.

**Software:** Thomas G. Scott, Michael J. Guertin.

**Supervision:** Daniel Gioeli, Michael J. Guertin.

**Validation:** Thomas G. Scott.

**Visualization:** Thomas G. Scott, Daniel Gioeli, Michael J. Guertin.

**Writing – original draft:** Thomas G. Scott.

**Writing – review & editing:** Thomas G. Scott, Kizhakke Mattada Sathyan, Daniel Gioeli, Michael J. Guertin.

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
