## [Decision Letter · Decision Letter 0]

27 Nov 2023

Dear Dr Guertin,

Thank you very much for submitting your Research Article entitled 'TRPS1 modulates chromatin accessibility to regulate estrogen receptor (ER) binding and ER target gene expression in luminal breast cancer cells' to PLOS Genetics.

The manuscript was fully evaluated at the editorial level and by independent peer reviewers. The reviewers appreciated the attention to an important problem, but raised some concerns about the current manuscript. Based on the reviews, we will not be able to accept this version of the manuscript, but we would be willing to review a revised version.

If you decide to revise the manuscript for further consideration at PLOS Genetics, please aim to resubmit within the next 60 days, unless it will take extra time to address the concerns of the reviewers, in which case we would appreciate an expected resubmission date by email to plosgenetics@plos.org.

We are sorry that we cannot be more positive about your manuscript at this stage. Please do not hesitate to contact us if you have any concerns or questions.

Yours sincerely,

Björn von Eyß

Guest Editor

PLOS Genetics

John Greally

Section Editor

PLOS Genetics

Reviewer's Responses to Questions

**Comments to the Authors:**

Reviewer #1: In this manuscript, Scott et al. evaluate the role of TRPS1 in ER+ breast cancer cells. Using public data, they find that TRPS1 is important for cell growth specifically in luminal (ER+) breast cancer cells. The authors then make a T-47D line harboring an inducible degron tag on TRPS1. With this model, they find that acute loss of TRPS1 leads to reduced TRPS1 genome binding, a direct increase in chromatin accessibility at several genomic loci, and an increase in expression of genes near TRPS1 binding sites. The authors also find that TRPS1 loss rewires ER genomic binding and impacts ER target genes. Finally, the authors show that the TRPS1 target gene signature in breast cancer samples is associated with worse patient outcomes. Overall, this manuscript reports an interesting finding in a rigorous manner. The data are well described and compelling on the whole. We believe that there are a few mostly minor changes that would improve the manuscript prior to publication.

1. The PRO-seq effect in Figure 3G is very subtle and does not appear to have been tested statistically. The authors should statistically test whether PRO-seq signal is reproducibly higher with TRPS1 depletion.

2. In the results and discussion, the authors refer to changes in ER binding or motif enrichment in ATAC sites as changes in ER activity. However, ER remains "active" and it would be more accurate to refer to this as changes in ER genomic binding or something similar.

3. Instead of "the estrogen receptor" it should be referred to as estrogen receptor alpha, including in the title and abstract.

4. For Figure 1C, the authors should clarify in the figure legend and x-axis that the black box/violin is for other cancer cell lines.

5. It would be helpful if the authors explained, possibly in the Discussion, why they chose T-47D when other breast cancer lines are more sensitive to TRPS1 KO based on Figure 1B.

6. Was the degron insertion heterozygous, the methods and results were not clear on this point? If so, why do the authors think that most of TRPS1 is removed?

7. The 24-hour timepoint is missing from Figure 3F.

8. Ideally, Figure 6C would include a control of parental cells treated with dTAG.

Reviewer #2: Scott et. al. used targeted protein degradation to investigate the primary effects of the GATA transcription factor TRPS1 in luminal breast cancer cells. Using a combination of TRPS1 Chip-seq, time-resolved ATAC-Seq and Pro-seq, the authors propose a model that TRPS1 directly represses the accessibility and transcription of its target genes. The authors describe that in addition to the GATA binding site, these promoters contain binding sites for the estrogen receptor (ER) and TRPS1 represses efficient binding of ER. In contrast, the authors propose an indirect activation mechanism: Genes that are downregulated after TRPS1 depletion (=TRPS1-activated) are not TRPS1 bound and are in principle accessible. Since these also contain ER binding sites, ER is bound in a TRPS1-proficient situation and transcription is active accordingly. After acute depletion, however, ER is sequestered to the previously TRPS1-bound promoters and is no longer available to the TRPS1-unbound promoters.

The manuscript is quite interesting, very clearly written and the experiments shown are valid and well presented. However, parts of the mechanism proposed here require further experimental confirmation.

Major points:

1. a crucial aspect of this study is the direct gene regulation by TRPS1. However, no gene regulation data have been obtained in intact cells (PRO-seq measures transcription after permiabilization). In my opinion, the authors should perform Slam-seq 30 minutes and 4 hours after dTAG and analyze the regulation of old and new RNA and compare it with the Pro-Seq data. Based on this, the authors should select a set of activated and repressed genes for qPCR analysis and use this assay to examine a number of luminal (ER-sensitive and ER-in-sensitive) and ER-negative breast cancer cell lines to show that the model is universal and that the gene regulatory program is truly ER-dependent.

2. the proposed indirect repression mechanism is plausible but unproven. The authors need to better substantiate this, e.g. by overexpression of ER and an associated attenuation of activation by TRPS1.

3. the Degron line is a great tool and should be used to phenotype acute TRPS1 loss cellularly (growth curve, cell cycle FACS, apoptosis, etc).

Elmar Wolf

Reviewer #3: This is a very interesting paper, clear and well written.

They chose to focus of TRPS1, a very important oncogene involved in luminal breast cancer, probably together with ER, but their functional relationships are still unclear.

They show that TRPS1 is associated with luminal breast cancer susceptibility and that luminal cancer cell lines are dependent on ER as well as on TRPS1 to proliferate.

The authors used a cutting-edge technology by integrating a degron in TRPS1 exon, so that they could induce its degradation within few minutes/hours with a chemical.

Nevertheless they also used PRO-seq to look at the real-time transcription activity.

They show that TRPS1 disappearance induces increase in chromatin accessibility together with relocation of ER at the chromatin.

The results are very convincing, all data and code are made available by the author.

This manuscript is another milestone in the road to the comprehension of ER/TRPS1 functional relations.

**Have all data underlying the figures and results presented in the manuscript been provided?**

Reviewer #1: Yes

Reviewer #2: Yes

Reviewer #3: Yes

PLOS authors have the option to publish the peer review history of their article (what does this mean?). If published, this will include your full peer review and any attached files.

Reviewer #1: No

Reviewer #2: **Yes: **Elmar Wolf

Reviewer #3: No

---

## [Decision Letter · Decision Letter 1]

30 Jan 2024

Dear Dr Guertin,

We are pleased to inform you that your manuscript entitled "TRPS1 modulates chromatin accessibility to regulate estrogen receptor alpha (ER) binding and ER target gene expression in luminal breast cancer cells" has been editorially accepted for publication in PLOS Genetics. Congratulations!

Yours sincerely,

Björn von Eyß

Guest Editor

PLOS Genetics

John Greally

Section Editor

PLOS Genetics

Comments from the reviewers (if applicable):

Reviewer's Responses to Questions

**Comments to the Authors:**

Reviewer #1: The authors have thoughtfully responded to my initial comments.

Reviewer #2: Although the authors have not addressed the points I raised through new experiments (but by specifying their assertions), I support the publication of this overall interesting and important manuscript.

Reviewer #3: Thank you for your revision

**Have all data underlying the figures and results presented in the manuscript been provided?**

Reviewer #1: None

Reviewer #2: Yes

Reviewer #3: None

PLOS authors have the option to publish the peer review history of their article (what does this mean?). If published, this will include your full peer review and any attached files.

Reviewer #1: No

Reviewer #2: **Yes: **Elmar Wolf

Reviewer #3: No

**Data Deposition**

http://datadryad.org/submit?journalID=pgenetics&manu=PGENETICS-D-23-01061R1

**Press Queries**

---

## [Editor Report · Acceptance letter]

10 Feb 2024

PGENETICS-D-23-01061R1 

TRPS1 modulates chromatin accessibility to regulate estrogen receptor alpha (ER) binding and ER target gene expression in luminal breast cancer cells 

Dear Dr Guertin, 

We are pleased to inform you that your manuscript entitled "TRPS1 modulates chromatin accessibility to regulate estrogen receptor alpha (ER) binding and ER target gene expression in luminal breast cancer cells" has been formally accepted for publication in PLOS Genetics! Your manuscript is now with our production department and you will be notified of the publication date in due course.

With kind regards,

Zsofia Freund

PLOS Genetics

On behalf of:
